# Oxidative Stress and Immune System Dysfunction in Autism Spectrum Disorders

**DOI:** 10.3390/ijms21093293

**Published:** 2020-05-06

**Authors:** Luca Pangrazzi, Luigi Balasco, Yuri Bozzi

**Affiliations:** Center for Mind/Brain Sciences (CIMeC), University of Trento, 38068 Rovereto, Italy; luigi.balasco@unitn.it (L.B.); yuri.bozzi@unitn.it (Y.B.)

**Keywords:** ROS, inflammation, ASD, autism, immune system

## Abstract

Autism Spectrum Disorders (ASDs) represent a group of neurodevelopmental disorders associated with social and behavioral impairments. Although dysfunctions in several signaling pathways have been associated with ASDs, very few molecules have been identified as potentially effective drug targets in the clinic. Classically, research in the ASD field has focused on the characterization of pathways involved in neural development and synaptic plasticity, which support the pathogenesis of this group of diseases. More recently, immune system dysfunctions have been observed in ASD. In addition, high levels of reactive oxygen species (ROS), which cause oxidative stress, are present in ASD patients. In this review, we will describe the major alterations in the expression of genes coding for enzymes involved in the ROS scavenging system, in both ASD patients and ASD mouse models. In addition, we will discuss, in the context of the most recent literature, the possibility that oxidative stress, inflammation and immune system dysfunction may be connected to, and altogether support, the pathogenesis and/or severity of ASD. Finally, we will discuss the possibility of novel treatments aimed at counteracting the interplay between ROS and inflammation in people with ASD.

## 1. Introduction

Autism Spectrum Disorders (ASD) form a heterogeneous group of neurodevelopmental syndromes characterized by persistent deficits in social communication and social interaction, and restricted, repetitive patterns of behavior, interests or activities [1]. Multiple conditions of comorbidity, such as anxiety and depression, attention-deficit/hyperactivity disorder (ADHD), obsessive-compulsive disorder (OCD), epilepsy and immune and autoimmune dysregulation, can be observed with ASD. The search for a molecular etiology is complicated by the interplay of both genetic and environmental factors that, together, contribute to ASD pathogenesis. Recently, advances in genomics and other molecular technologies have enabled the study of ASD at the molecular level, illuminating genes and pathways whose perturbations help to explain the clinical variability among patients and whose impairments provide possible opportunities for better treatment options. Despite this, up to now, no treatment is available for ASD. 

Neurobiological research on ASD has traditionally focused on the pathways involved in neural development and synaptic plasticity. However, several lines of evidence suggest that immune dysregulation may lead to, or at least contribute to, ASD. In addition to this, the role of oxidative stress in supporting the pathogenesis of this group of diseases has been reported in several studies. Interestingly, a connection between the accumulation of oxygen radicals and immune dysfunction may exist.

Animal research has rapidly advanced in recent years, and numerous models displaying many of the features characteristic of autism have been suggested (https://gene.sfari.org/). According to the AutDB database (http://autism.mindspec.org/autdb/Welcome.do, updated January 2020), 3145 animal models of ASD, including inbred, induced and genetic mouse models, are currently available. Genetic studies demonstrated that mutations in several genes coding for synaptic proteins, such as SHANK3 [2], NLGN3, NLGN4X [3], CNTNAP2 [4] and GABRB3 [5], are associated with ASD. Furthermore, ASD is syndromic with other neuropsychiatric conditions with single gene mutations including Fragile X syndrome (FMR1) [6], tuberous sclerosis (either TSC1 or TSC2) [7], Cowden syndrome (PTEN) [8] and Angelman syndrome (UBE3A) [9]. Interestingly, several genes associated with ASD, such as PTEN, TSC1 and TSC2, all involved in the phosphoinositide-3-kinase (PI3K) pathway, display immunoregulatory functions. 

In this review, we first summarize recent literature discussing the contribution of oxidative stress to ASD. Taking advantage of a meta-analysis performed using the database dbMDEGA, we describe how genes involved in the ROS scavenging system are expressed in both ASD patients and mouse models of ASD. In addition, we will discuss about how oxidative stress may be linked to neuroinflammation, therefore contributing to an ASD-like phenotype. Finally, we will summarize the results of some studies, in which interventions using antioxidants as supplements or included in foods led to improvements in ASD symptoms.

## 2. The Contribution of Oxidative Stress to ASD

The imbalance between the synthesis of reactive oxygen/nitrogen species (ROS/RNS) and the organism’s ability to block their deleterious effects using antioxidative systems is the cause of oxidative stress. Mitochondrial abnormalities and oxidative stress play causative roles in the aging process and are frequently present in cases of diseases [10,11]. Mitochondrial dysfunction has been associated with the accumulation of ROS within the cells, which, when not properly scavenged, may lead to oxidative stress [12]. In order to counteract this condition, several mechanisms are required for the detoxification from toxic compounds and the neutralization of oxygen/nitrogen radicals that are present in every cell (Figure 1). Superoxide (O_2_^−^) can be formed as a byproduct of the normal metabolism of oxygen. Although this molecule, similarly to the other oxygen species, may play an important role in cell signaling and homeostasis [13], its accumulation may cause damage to cell structures, therefore supporting oxidative stress. For this reason, superoxide is immediately converted to hydrogen peroxide (H_2_O_2_) by a class of enzymes known as superoxide dismutases (SODs). The presence of H_2_O_2_ may be toxic for the cells, as it can pass through cell membranes and damages the DNA. Thus, several neutralization pathways have been developed for scavenging hydrogen peroxide. Among them, the most important enzymes are catalase and glutathione peroxidase (GPx), which both convert H_2_O_2_ to H_2_O. The tripeptide glutathione is one of the most important detoxifying agents, which plays a fundamental role in scavenging ROS. In its reduced form (GSH), glutathione donates an electron to H_2_O_2_ in a reaction catalyzed by GPx, and it is converted to the oxidized form. GSH can be re-generated again by glutathione reductase, which utilizes NAD(P)H as the electron donor. Glutathione can also act as a cofactor for other enzymes such as GSH transferase, helping in removing toxic molecules from the cells. The brain represents one of the major metabolizers of oxygen, and therefore, large amounts of ROS accumulate within several brain regions. Despite this, at least in some conditions, relatively weak protective mechanisms are present. For this reason, the brain may be very sensitive to the assaults caused by the accumulation of radicals (as summarized by [14]). 

### 2.1. Oxidative Stress Is Elevated in ASD Patients

The production of ROS represents a key feature of many, if not all, neurological disorders. Specifically, oxygen radicals have been shown to play a fundamental role in the pathogenesis, progression and severity of Alzheimer’s, Parkinson’s and Huntington’s diseases, as well as autism and amyotrophic lateral sclerosis [15,16,17,18]. In addition, ASD patients are more prone to undergoing oxidative stress and are very vulnerable to ROS-mediated damage and neuronal toxicity [19,20]. Thus, several studies have now linked ASD to increased ROS levels and reduced antioxidant capacity, not only in the brain, but also at the systemic level. Some reports have investigated molecules related to oxidative stress in the brains of people with ASD [21,22]. Interestingly, oxidative stress characterizes not only individuals with ASD but, in some cases, also their parents. Impairments in the metabolism of glutathione have additionally been documented. In particular, increased levels of oxidized glutathione, low reduced glutathione and diminished glutathione redox ratios were measured in both the temporal cortices and cerebella of autistic patients [22].

Other groups have investigated parameters related to oxidative stress in the periphery, as potential biomarkers for the early diagnosis of ASD. SOD activity in erythrocytes was higher in ASD children in comparison with the controls [23]. An increased SOD level is considered a compensatory mechanism to counteract the cell damage caused by oxidative stress in the brain. In addition, catalase activity was reduced in erythrocytes but did not change inside the plasma. The plasma concentrations of reduced GSH were found to be lower and the ratio of oxidized GSH/reduced GSH was higher in autistic patients compared to in healthy controls [21]. In addition, GSH in the plasma of autistic children was lower when compared with the controls [24]. Furthermore, González-Fraguela and co-workers showed how markers of damage to biomolecules, such as malonyldialdehyde (MDA) and 8–hydroxy-2 deoxyguanosine (8OHdG), and GSH levels were reduced with autism [25]. In accordance with the results of Rose et al., a reduced antioxidant capacity—i.e., altered SOD, GPx and CAT activities—were found in autistic persons when compared to controls. More recently, the molecular basis of the pathophysiological role of ROS in autism has been assessed in an Egyptian cohort [26]. The transcriptional pattern of 84 genes related to oxidative stress was measured in peripheral blood mononuclear cells (PBMCs) isolated from autistic patients and healthy controls. Eight genes coding for key proteins involved in the metabolism of ROS (GCLM, SOD2, NCF2, PRNP, PTGS2, TXN and FTH1) were downregulated in autistic persons. These molecules may be useful as biomarkers of autism, being relevant for early diagnostic and therapeutic purposes. In addition to this, in ASD patients, red blood cells (RBCs) are known to be damaged by ROS [27]. Indeed, in these individuals, oxygen radicals alter RBC shape and morphology, which therefore differ from those in the healthy controls. In addition, chronic mitochondrial dysfunctions associated with electron transport chain (ETC) complex I and III have been identified in patients with ASD [28,29]. 

### 2.2. Oxidative Stress in Human ASD Samples and Mouse Models: A Meta-Analysis

A well-structured meta-analysis of brain gene expression profiles from currently available expression datasets, collected in both human ASD patients and ASD mouse models, is available in the dbMDEGA database (https://dbmdega.shinyapps.io/dbMDEGA). In particular, the expression analysis performed in different brain areas was obtained from three human ASD brain studies (137 samples in total) and from 14 mouse models of ASD. When we interrogated the database for the most important genes coding for enzymes involved in the ROS scavenging system (Figure 1), the expression of many genes was significantly reduced in ASD patients, in comparison to in the healthy controls (Table 1). It is important to stress that the samples included in the meta-analysis were very heterogeneous and were obtained from different brain areas. Thus, if significant differences can be identified in this context, we can speculate that the results might be even more interesting if the analysis was to be performed with more details. Despite this, these results suggest that oxidative stress may be elevated in ASD, in accordance with the studies described in Section 2.1. 

The expression of genes coding for enzymes involved in the ROS scavenging system is also known to be altered in mouse models of ASD. BTBR T+tf/J (BTBR) mice show peripheral and CNS abnormalities similar to autistic patients, and therefore, they have extensively been used as a mouse model of ASD [30]. In BTBR mice, lower levels of glutathione and the enzymatic antioxidants SOD and glutathione peroxidase can be observed in the cerebellum and peripheral immune cells [31]. 

When we analyzed the expression of molecules involved in the ROS scavenging system in ASD mouse models present in the dbMDEGA database, similar results could be observed. In particular, the levels of SOD2 and GPX3, two of the genes differentially expressed in ASD patients, were also reduced in at least some of the ASD models (Figure 2). 

In summary, strong evidence exists that oxidative stress may represent a major contributor to autistic-like behavior in individuals with ASD and ASD mouse models.

## 3. Immune System Dysfunction in ASD 

The immune system has been classically divided into two main branches, innate and adaptive immunity. While the responses generated by innate immune cells are more non-specific and recognize conserved structures expressed by pathogens, adaptive immune cells can develop antigen-specific responses, which give rise to immunological memory. In the brain, the most frequent cell type is the microglia, which belong to the innate immune system and represents 80% of the overall amount of brain immune cells [32] and 10%–15% of total brain cells [33] Microglia act as brain-resident macrophages, which play a fundamental role in protecting the functions of this organ. Indeed, microglia do not only scavenge damaged neurons and synapses within the central nervous system (CNS) but can also block infectious agents when they cross the blood–brain barrier. Other subpopulations that can be identified in the brain include innate immune cells such as monocytes, neutrophils, dendritic cells and natural killer (NK) cells, and adaptive immune cells like B cells and T cells, which, altogether, are known as lymphocytes [32,34]. Although lymphocytes are scarce within the brain, their importance in supporting brain functions such as synaptic plasticity has been documented [35].

The first link between immune system dysfunction and ASD was proposed for the first time over 40 years ago [36]. Since then, research has looked more closely at the potential contribution of impairments in the immune system to ASD. More recently, studies performed in ASD patients indicated that immune system dysfunction is often supported by a strong inflammatory state [37,38,39]. In particular, increased levels of pro-inflammatory cytokines interleukin (IL)-1β, IL-6, IL-8, IL-12p40, tumour necrosis factor (TNF) and IL-17 were found in the plasma of 97 medication-free ASD children [40,41].

A key finding showing the presence of impairments in immune system function was the observation of the enhanced expression of neuroinflammation markers in post-mortem samples from ASD individuals. These studies detected signs of microglia activation as well as increased inflammatory cytokines and chemokines (i.e., interferon (IFN)γ, IL-1β, IL-6, TNF and chemokine C-C motif ligand (CCL)-2) in the brains and cerebrospinal fluid of ASD subjects [42,43,44]. Dysfunctions in the central and peripheral immune systems of ASD individuals have also been described. These include the over-stimulation of immune cells, a cytokine/chemokine imbalance, and an increased permeability of the blood–brain barrier [45,46,47,48,49,50,51,52]. Several studies have proposed a possible correlation between the autistic phenotype and different aspects of altered immunity, such as cytokine levels and secretion [53], the levels of immunoglobulins (IgM and IgG) [54], B lymphocyte antigen D8/17 expression [55], serum antineuronal antibodies [56,57], M1 antibodies [58] and maternal antibody status [59]. Altogether, these data support the hypothesis that immune dysregulation may contribute, either fully or in part, to the autistic phenotype. Epidemiological studies showed increased rates of immune dysregulation in both mothers and fathers of children with ASD [60,61,62,63,64]. Therefore, research investigating these aspects is aimed at developing animal models mimicking this risk factor. These include maternal respiratory infection with influenza virus and maternal immune activation (MIA) with either polyinosine:cytosine (poly(I:C), a synthetic, double-stranded RNA that evokes an antiviral-like immune reaction) or lipopolysaccharide (LPS), which evokes an antibacterial-like immune reaction [65]. As an example, the pro-inflammatory cytokine IL-6 appears to play a key role in all the above mentioned models, and exposure to IL-6 alone during gestation was reported to be sufficient to elicit behavioral changes in the offspring [66,67].

Taken together, these results suggest that the increased production of pro-inflammatory molecules in ASD may represent a strong contributor to the pathogenesis and the severity of these disorders.

## 4. Infiltration of Immune Cells in the Brain: the Link between ROS, Inflammation and Neurodegeneration 

Whether high levels of ROS may be the cause of inflammatory conditions or whether inflammation may induce oxidative stress is unclear. Despite this, a relationship between oxidative stress and inflammation has been documented, as evidence indicated that oxidative stress supports chronic inflammatory diseases. In several cell populations, high concentrations of ROS can activate signaling pathways and create vicious cycles, which maintain a high secretion of pro-inflammatory cytokines and chemokines [68,69]. In the bone marrow (BM) from elderly persons, in which ROS levels were shown to be particularly high, pro-inflammatory molecules such as IFNγ increase ROS levels within BM cells [70,71]. This supports the production of IL-15 and IL-6, two additional inflammatory mediators, which maintain other pro-inflammatory cell types such as T cells and NK cells within the BM. In this way, a vicious cycle of inflammation–ROS–inflammation may take place in the BM, leading to an impairment in the functionality of this organ as a result [72]. A similar situation may exist also with brain diseases. In this context, immune cells such as microglia cells and T cells infiltrated within the brain may secrete pro-inflammatory molecules, which support the above-mentioned vicious cycle of inflammation–oxidative stress–inflammation [14]. 

In the presence of neurological disorders such as ASD, oxidized proteins and lipid peroxidation, both caused by the accumulation of ROS, may directly induce neuroinflammation. This situation may cause cell death, therefore leading to neuron degeneration [14]. In addition, protein oxidation may induce the release of peroxiredoxin 2 (PRDX2), a known inflammatory signal acting as a redox-dependent inflammatory mediator that activates macrophages to produce TNF [73]. Furthermore, lower levels of GSH lead to higher ROS production, which directly promotes inflammation [74]. When pathological conditions such as diabetes are present, oxidative stress can increase the levels of pro-inflammatory cytokines TNF and IL-6, vascular cell adhesion molecule-1 (VCAM-1), intercellular adhesion molecule-1 (ICAM-1) and nuclear factor-kappa B (NF-κB), which, altogether, support neuron degeneration, resulting in diabetic encephalopathy [75]. Thus, it is evident that when oxidative stress develops in one organ, it can easily turn into inflammation and propagate into the brain, therefore causing neurodegeneration. 

Microglia are known to play an important role in supporting brain inflammation. With their motile protrusion, microglia continuously explore the brain microenvironment in order to detect the presence of viral and bacterial infections, toxins and local tissue injury [76]. In addition, it is now evident that microglia are fundamental for typical CNS development and function, such as the phagocytosing apoptotic neurons, providing trophic support to developing neurons and vasculature and helping in synaptic pruning [77,78,79]. As this cell population is involved in a broad spectrum of functions, microglia cells need to be very plastic in order to easily change their phenotype in response to different stimuli [76,80]. In particular, microglia cells can easily switch from a non-inflammatory phenotype, when they are implicated in the regulation of brain functions, to a pro-inflammatory role, when stress conditions such as infections or oxygen radicals are present in the brain. Importantly, microenvironmental factors are known to define and modulate the identity of microglia cells. In several studies, neuroinflammation has been linked to the presence of activated microglia cells within the brain. Thus, with neurological disorders such as ASD, microglia may become altered, therefore promoting neuroinflammation. Indeed, the activation of these cells has been documented in individuals with ASD, and this has been associated with a loss of connections or underconnectivity in their brains [81].

Several studies have described how microglia and astrocytes may produce several inflammatory mediators in response to oxidative stress [82,83,84]. In addition to this, brain-infiltrated T cells may play an important role in neuroinflammation [85,86]. Very recently, an accumulation of IFNγ-producing T cells was observed in proximity to the neural stem cells (NSCs) of old mice, in comparison with young controls [87]. Interestingly, IFNγ was able to inhibit the proliferation of NSCs. Thus, these results suggest that pro-inflammatory molecules, secreted either by microglia cells or T cells, may directly impair brain functions, most probably in a ROS-mediated manner. Whether a similar situation may be present in the brain with ASD has not been investigated yet.

## 5. Targeting ROS to Treat ASD 

As oxidative stress seems to represent an important feature of ASD patients, many studies have investigated whether ASD-like behaviors may be improved after treatment with antioxidants. These approaches are based on the fact that, as reported in Section 2.1, the expression of antioxidant enzymes is reduced in ASD. In addition, the levels of endogenous antioxidant molecules are known to be reduced in ASD individuals in comparison to in healthy controls [88]. For these reasons, the administration of external antioxidants may help in boosting the endogenous ROS scavenging system, therefore counteracting oxidative stress. In order to be optimal therapeutic candidates for ASD, antioxidants must cross the blood–brain barrier (BBB) and enter the brain parenchyma, in which they must reach their optimal therapeutic concentrations. 

Coenzyme Q10 (CoQ10) (ubiquinone) is a mitochondrial antioxidant cofactor that crosses the BBB. In children with ASD, the administration of ubiquinol (the reduced form of coenzyme Q10) led to an improvement in communication with parents and in overall verbal communication [89]. In a more recent study, coenzyme Q10 supplementation was shown to reduce oxidative stress and ASD-like symptoms in ASD children [90]. Another promising molecule is N-acetylcysteine (NAC), a strong antioxidant that helps in boosting glutathione levels [91]. In the synthesis of the tripeptide glutathione, the limiting step is represented by the addition of the amino acid cysteine, levels of which are generally low within the cells, particularly in the presence of diseases. In this context, NAC is considered an important source of cysteine, and it becomes even more powerful in conditions in which the levels of glutathione are very low. The effects of NAC on ASD-like behaviors were investigated in rats, in a valproic acid (VPA)-induced model of autism. The authors reported that, in these animals, NAC ameliorated repetitive and stereotypic activity and reduced the levels of oxidative stress by increasing glutathione and reducing malondialdehyde levels in comparison with the controls [92]. Benefits could be observed also when NAC was administered to children suffering from ASD. In particular, a study showed that the drug was well tolerated by the patients and could significantly reduce irritability [93]. Furthermore, other studies reported that stereotypical behaviors could be improved after 8 weeks of treatment with NAC [94,95]. Another antioxidant tested in the context of ASD is vitamin C. When this molecule was supplemented to ASD children for 30 weeks, a significant improvement in sensorimotor behaviors could be observed [96,97]. Many double-blind, placebo-controlled clinical trials using combinations of antioxidants such as vitamin C, carnosine, zinc, vitamin B6 and magnesium have been performed (https://clinicaltrials.gov/). Treatments with high doses of vitamin C or carnosine, and the combination of vitamin B6 and magnesium, improved the behavior of individuals with ASD. Moreover, magnesium deficiency has been also associated with oxidative stress [98]. In ASD patients, typically characterized by low levels of magnesium in the blood, magnesium supplementation leads to improvements of symptoms, such as poor concentration and hyperactivity [99]. In addition, it was shown that in valproic acid (VPA)-exposed dams, treatment with high doses of folic acid, vitamin E and the methyl donor methionine could ameliorate or prevent most of the VPA-induced damage [100,101]. S-adenosylmethionine (SAM) is known to reduce oxidative stress in several organs [102,103,104]. In the brain, this molecule inhibited lipid peroxidation and supported the glutathione system [102]. In 60 day old mice exposed to VPA, the co-administration of SAM decreased VPA-induced oxidative stress in the brain and improved ASD-like behaviors [101]. The hormone melatonin was additionally shown to act as a potent antioxidant. In several studies, melatonin treatment improved daytime behavior in individuals with ASD [105]. It is important to underline that many of these molecules can be found at high concentrations in various plants, fruits and foods. For this reason, in the last few years, numerous nutritional interventions have been planned for ASD patients, with the aim of improving the lives of these individuals. As an example, a recent study showed that the consumption of high concentrations of the antioxidant cacao, which is present in the dark chocolate, could improve the social communication, unusual behaviors and self-regulation behaviors of children with ASD [106]. Thus, it is becoming evident that antioxidants and vitamins, taken as supplements or included in foods, may strongly help ASD persons in attenuating their ASD-associated symptoms.

## 6. Conclusions

It is becoming clearer how immune system dysfunctions may be associated with ASD. Indeed, impairments in both the innate and adaptive immune system support the onset of pro-inflammatory conditions, which may then lead to oxidative stress. In parallel, it is possible that the metabolism of several cell types (particularly brain cells) may be altered in ASD. In this way, oxidative stress may be promoted, which can then directly induce chronic inflammatory conditions. Thus, a strong connection between the immune system, inflammation and oxidative stress must be expected in ASD individuals. This opens up the possibility for novel treatments aiming at attenuating ASD-like behaviors in the patients. Many studies are currently investigating whether selected antioxidants present in plants, fruits and other foods may lead to improvements in the patients. In addition, dietary interventions are currently being planned in people with ASD. As oxidative stress and inflammation are strongly inter-connected, supplementation with anti-inflammatory drugs may additionally help in reducing the levels of oxygen radicals in the patients. As a proof of concept, the use of anti-inflammatory drugs has shown benefits in individuals with ASD [107]. Novel natural antioxidant and anti-inflammatory compounds must be tested in the clinic, in order to provide new weapons to fight against ROS/inflammation and immune system dysfunction in ASD in the best way. In addition, new strategies that may consider oxidative stress as a biomarker for the diagnosis and prognosis of ASD must be implemented.

## Figures and Tables

**Figure 1 ijms-21-03293-f001:**
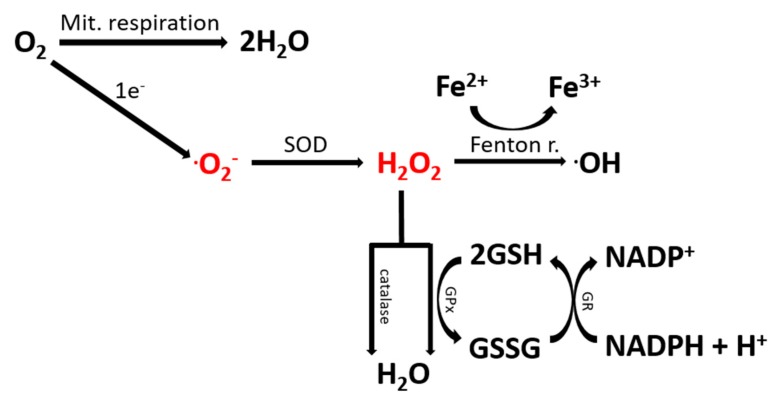
Reactions required for the detoxification from reactive oxygen species (ROS). Enzymes catalyzing the major detoxification reactions are reported. Abbreviations: Fenton r.: Fenton reaction; GPx: glutathione peroxidase; GR: glutathione reductase; Mit.: mitochondrial; SOD: superoxide dismutase.

**Figure 2 ijms-21-03293-f002:**
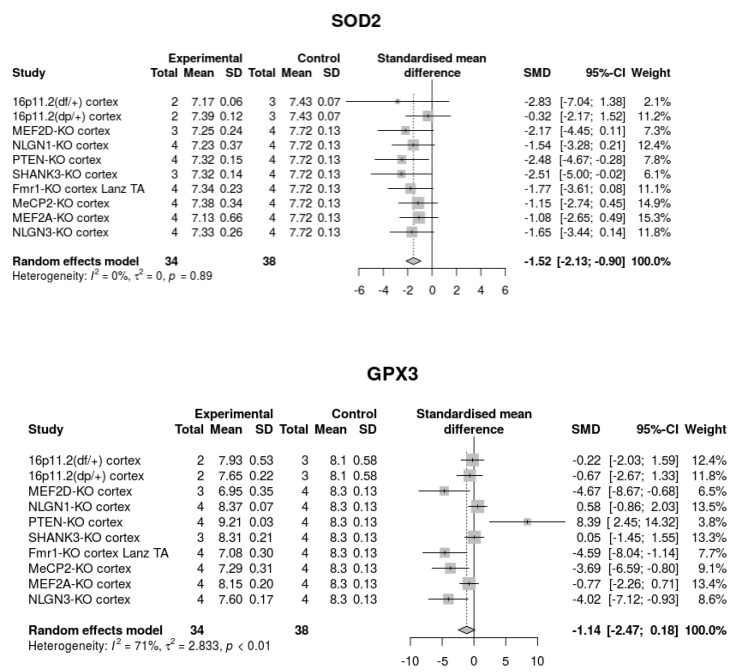
Genes coding for enzymes involved in the ROS scavenging system are differentially expressed in mouse models of ASD. For simplicity, the gene expression of only two genes (SOD2 and GPX3) in nine out of 14 mouse models of ASD are shown. The complete information is available in the dbMDEGA database.

**Table 1 ijms-21-03293-t001:** Genes coding for enzymes involved in the ROS scavenging system are differentially expressed in autistic spectrum disorder (ASD) patients. Differences in the expression of genes involved in the ROS scavenging system (Figure 1), after the comparison between human ASD patients and healthy controls, calculated by the dbMDEGA database. Genes of interest were submitted in the “Meta_Summary” section of the database and meta-analysis was performed using the tau-squared test. Tau-squared values, *p* values and false discovery rates (FDRs) obtained from statistical analysis are reported for each gene. A *p* value < 0.05 and an FDR < 0.25 are considered significant. Significant genes are underlined and reported in bold within the table.

Gene	Statistic Test	*p*_Value	FDR	Expression in ASD Mouse Models
**SOD1**	1.24	0.11	0.26	no change
**SOD2**	**3.47**	**0.00**	**0.03**	**down**
**SOD3**	**1.68**	**0.05**	**0.19**	**down**
**CAT**	0.60	0.27	0.39	no change
**GPX1**	0.31	0.38	0.45	no change
**GPX2**	0.05	0.48	0.49	no change
**GPX3**	**2.57**	**0.01**	**0.08**	**down**
**GPX4**	**1.69**	**0.05**	**0.19**	**down**
**GSTM1**	1.19	0.12	0.27	no change
**GSR**	0.75	0.23	0.36	no change
**GSTA1**	**2.68**	**0.00**	**0.07**	**up**
**GSTA4**	**2.46**	**0.01**	**0.09**	**down**

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
