# Peer review of "Oxidative Stress and Immune System Dysfunction in Autism Spectrum Disorders"

_ijms, 2020, doi:10.3390/ijms21093293_

Round 1
Reviewer 1 Report
The authors have carried out a review of oxidative stress and also neuroinflammation in ASD. This review was easy to read and the English was fairly good except for the term "evidences' which should be changed to evidence in a few places. This review covered basic information with some new data from a Megabase on protein levels involved with ROS metabolism that was low in ASD. Overall this review adds to the field of ASD and ROS involvement in inflammation.
Author Response
The authors have carried out a review of oxidative stress and also neuroinflammation in ASD. This review was easy to read and the English was fairly good except for the term "evidences' which should be changed to evidence in a few places. This review covered basic information with some new data from a Megabase on protein levels involved with ROS metabolism that was low in ASD. Overall this review adds to the field of ASD and ROS involvement in inflammation.
We thank the reviewer for the suggestion. The word “evidences” has now been substituted with “evidence”.
Reviewer 2 Report
The present article by Pangrazzi et al. entitled:
“Oxidative stress and immune system dysfunction in autism spectrum disorders”
is a very comprehensive description of the subject.
It reviews available information on the mechanism related to oxidative stress that appears impaired in autistic patients, as well as on the treatments using antioxidants for improving autistic behaviors. It also provides data on the abnormal levels of genes encoding enzymes involved in the ROS scavenging system extracted from dbMDEGA database on mice models of ASD and ASD patients.
I believe it is a well-written and documented review and I would highly recommend it for publication, because it is a good synthesis of the effect of oxidative stress in ASD, well related to the inflammatory mechanisms in the brain, and also informs on the antioxidant treatments that could be useful to help autistic patients.
As minor suggestions I would recommend authors:
- Line 43: you may want to add Sfari hyperlink containing an updated list of ASD-related genes.
- 68: please add “-“ to O2- , so it is superoxide and not oxygen.
- 127: oxidative “stress”.
- 179: “microglia acts”… instead of Microglia are composed by
- 262: This has been associated to loss of connections or underconnectivity in “their” brain. of these people
I would also suggest authors to include information on other antioxidants such as SAMe and melatonin:
VPA exposed dams treated with high doses of folic acid and vitamin E (antioxidants) and the methyl donor methionine, ameliorated or prevented most VPA-induced damage [1, 2].
SAMe has also been shown to reduce oxidative stress [3-6], it inhibits lipid peroxidation and enhance glutathione system in the brain [3]. In 60 day old mice exposed to VPA on postnatal day 4 co-administration of SAMe significantly improved ASD like behavior and reduced the brain oxidative stress induced by VPA [2].
Hormone melatonin acts also as a potent antioxidant. Rossignol and Frye 2011 reviewed several studies in which melatonin treatments improved daytime behavior in ASD.
- Ehlers, K., M.M. Elmazar, and H. Nau, Methionine reduces the valproic acid-induced spina bifida rate in mice without altering valproic acid kinetics. J Nutr, 1996. 126(1): p. 67-75.
- Ornoy, A., et al., S-adenosyl methionine prevents ASD like behaviors triggered by early postnatal valproic acid exposure in very young mice. Neurotoxicol Teratol, 2018.
- Villalobos, M.A., et al., Effect of S-adenosyl-L-methionine on rat brain oxidative stress damage in a combined model of permanent focal ischemia and global ischemia-reperfusion. Brain Res, 2000. 883(1): p. 31-40.
- Gonzalez-Correa, J.A., et al., Effects of S-adenosyl-L-methionine on hepatic and renal oxidative stress in an experimental model of acute biliary obstruction in rats. Hepatology, 1997. 26(1): p. 121-7.
- Li, Q., et al., S-Adenosylmethionine Attenuates Oxidative Stress and Neuroinflammation Induced by Amyloid-beta Through Modulation of Glutathione Metabolism. J Alzheimers Dis, 2017. 58(2): p. 549-558.
- Yoon, S.Y., et al., S-adenosylmethionine reduces airway inflammation and fibrosis in a murine model of chronic severe asthma via suppression of oxidative stress. Exp Mol Med, 2016. 48(6): p. e236.
- Rossignol and Frye 2011. Melatonin in autism spectrum disorders: a systematic review and meta-analysis.
Author Response
The present article by Pangrazzi et al. entitled:
“Oxidative stress and immune system dysfunction in autism spectrum disorders”
is a very comprehensive description of the subject.
It reviews available information on the mechanism related to oxidative stress that appears impaired in autistic patients, as well as on the treatments using antioxidants for improving autistic behaviors. It also provides data on the abnormal levels of genes encoding enzymes involved in the ROS scavenging system extracted from dbMDEGA database on mice models of ASD and ASD patients.
I believe it is a well-written and documented review and I would highly recommend it for publication, because it is a good synthesis of the effect of oxidative stress in ASD, well related to the inflammatory mechanisms in the brain, and also informs on the antioxidant treatments that could be useful to help autistic patients.
As minor suggestions I would recommend authors:
Line 43: you may want to add Sfari hyperlink containing an updated list of ASD-related genes.
Thank you for the suggestion, we now added the Sfari hyperlink.
68: please add “-“ to O2- , so it is superoxide and not oxygen.
127: oxidative “stress”.
179: “microglia acts”… instead of Microglia are composed by
262: This has been associated to loss of connections or underconnectivity in “their” brain. of these people
The text has been modified according to the suggestions of the reviewer.
I would also suggest authors to include information on other antioxidants such as SAMe and melatonin:
VPA exposed dams treated with high doses of folic acid and vitamin E (antioxidants) and the methyl donor methionine, ameliorated or prevented most VPA-induced damage [1, 2].
SAMe has also been shown to reduce oxidative stress [3-6], it inhibits lipid peroxidation and enhance glutathione system in the brain [3]. In 60 day old mice exposed to VPA on postnatal day 4 co-administration of SAMe significantly improved ASD like behavior and reduced the brain oxidative stress induced by VPA [2].
Hormone melatonin acts also as a potent antioxidant. Rossignol and Frye 2011 reviewed several studies in which melatonin treatments improved daytime behavior in ASD.
- Ehlers, K., M.M. Elmazar, and H. Nau, Methionine reduces the valproic acid-induced spina bifida rate in mice without altering valproic acid kinetics. J Nutr, 1996. 126(1): p. 67-75.
- Ornoy, A., et al., S-adenosyl methionine prevents ASD like behaviors triggered by early postnatal valproic acid exposure in very young mice. Neurotoxicol Teratol, 2018.
- Villalobos, M.A., et al., Effect of S-adenosyl-L-methionine on rat brain oxidative stress damage in a combined model of permanent focal ischemia and global ischemia-reperfusion. Brain Res, 2000. 883(1): p. 31-40.
- Gonzalez-Correa, J.A., et al., Effects of S-adenosyl-L-methionine on hepatic and renal oxidative stress in an experimental model of acute biliary obstruction in rats. Hepatology, 1997. 26(1): p. 121-7.
- Li, Q., et al., S-Adenosylmethionine Attenuates Oxidative Stress and Neuroinflammation Induced by Amyloid-beta Through Modulation of Glutathione Metabolism. J Alzheimers Dis, 2017. 58(2): p. 549-558.
- Yoon, S.Y., et al., S-adenosylmethionine reduces airway inflammation and fibrosis in a murine model of chronic severe asthma via suppression of oxidative stress. Exp Mol Med, 2016. 48(6): p. e236.
Rossignol and Frye 2011. Melatonin in autism spectrum disorders: a systematic review and meta-analysis.
We thank the reviewer for this valuable suggestion. We added these references in the manuscript and discussed about the role of SAM and melatonin in ASD.